# Rapid Screening and Comparison of Chimeric Lysins for Antibacterial Activity against *Staphylococcus aureus* Strains

**DOI:** 10.3390/antibiotics12040667

**Published:** 2023-03-29

**Authors:** Jin-Mi Park, Dae-Sung Ko, Hee-Soo Kim, Nam-Hyung Kim, Eun-Kyoung Kim, Young-Hye Roh, Danil Kim, Jae-Hong Kim, Kang-Seuk Choi, Hyuk-Joon Kwon

**Affiliations:** 1Laboratory of Poultry Medicine, College of Veterinary Medicine, Seoul National University, Seoul 08826, Republic of Korea; 2Research Institute for Veterinary Science, College of Veterinary Medicine, BK21 for Veterinary Science, Seoul 08826, Republic of Korea; 3Department of Farm Animal Medicine, College of Veterinary Medicine, Seoul National University, Pyeongchang-gun 25354, Republic of Korea; 4Laboratory of Avian Diseases, College of Veterinary Medicine, Seoul National University, Seoul 08826, Republic of Korea

**Keywords:** *Staphylococcus aureus*, chimeric lysin, screening test method, antibacterial activity, autolysin, cell-free expression system

## Abstract

Chimeric lysins composed of various combinations of cell wall-lysing (enzymatic) and cell-wall-binding (CWB) domains of endolysins, autolysins, and bacteriocins have been developed as alternatives to or adjuvants of conventional antibiotics. The screening of multiple chimeric lysin candidates for activity via *E. coli* expression is not cost effective, and we previously reported on a simple cell-free expression system as an alternative. In this study, we sufficiently improved upon this cell-free expression system for use in screening activity via a turbidity reduction test, which is more appropriate than a colony reduction test when applied in multiple screening. Using the improved protocol, we screened and compared the antibacterial activity of chimeric lysin candidates and verified the relatively strong activity associated with the CHAP (cysteine, histidine-dependent amidohydrolase/peptidase) domain of secretory antigen SsaA-like protein (ALS2). ALS2 expressed in *E. coli* showed two major bands, and the smaller one (subprotein) was shown to be expressed by an innate downstream promoter and start codon (ATG). The introduction of synonymous mutations in the promoter resulted in clearly reduced expression of the subprotein, whereas missense mutations in the start codon abolished antibacterial activity as well as subprotein production. Interestingly, most of the *S. aureus* strains responsible for bovine mastitis were susceptible to ALS2, but those from human and chicken were less susceptible. Thus, the simple and rapid screening method can be applied to select functional chimeric lysins and define mutations affecting antibacterial activity, and ALS2 may be useful in itself and as a lead molecule to control bovine mastitis.

## 1. Introduction

*Staphylococcus aureus (S. aureus)* is a zoonotic/reverse zoonotic pathogen that causes food poisoning in humans, mastitis in cows, and arthritis in poultry [1,2,3]. Since the first discovery of methicillin-resistant *S. aureus* in 1961, the multi-drug resistance (MDR) of *S. aureus* has become a real threat to public health and the farm animal industry [4,5,6]. Humans are a natural host of *S. aureus*, and *S. aureus* from humans have adapted to other animals as a result of spillover [7,8,9,10]. Multilocus sequence typing (MLST) and pulsed-field gel electrophoresis (PFGE) have been successfully applied in molecular epidemiological studies, and a more cost-effective and clinically informative method, *rpoB* sequence typing (RSTing), has been established [11]. Recently, more extensive experimental and in silico RSTing that included *S. aureus* strains from Korean bovine mastitis and the GenBank database has demonstrated the apparent prevalence of RST4-1 and RST2-1 in humans worldwide in addition to the presence of RST10-2 and RST3-1 in Korean mastitis-suffering cows and diseased chickens, respectively [11,12,13].

The turgor pressure of *S. aureus* is 20–25 atm, and the maintenance of cell wall integrity is essential for survival [14]. Peptidoglycan is the major component of and provides stiffness for the cell wall [15]. *S. aureus* possesses a thick peptidoglycan layer of about 19 and 32 nm thickness for live and extracted cell walls, respectively [16]. As with other gram-positive bacteria, *S. aureus* peptidoglycan mesh is composed of glycan strands of 18 repeated disaccharides (*N*-acetyl muramic acid (NAM) and *N*-acetyl glucosamine (NAG)) on average and pentaglycine bridges cross-linking stem peptides bound to NAM of repeating disaccharides [17]. A very high percentage of up to 90% of stem peptides are cross-linked, which renders *S. aureus* peptidoglycan with high stiffness [15]. MDR *S. aureus* in humans and farm animals has encouraged the development of alternative antibiotics aimed at attacking peptidoglycan integrity. Endolysins are peptidoglycan-lysing enzymes and expressed at the end of the lytic cycle of bacteriophages for release of assembled progeny. Exogenous treatment has demonstrated that recombinant endolysins are novel antimicrobials, and various recombinant endolysins are being assessed in clinical trials [18,19,20]. Modular endolysins are composed of various enzymatic domains and cell-wall-binding domains that have independent activities [21]. LysK of bacteriophage K is one of the first staphylococcal endolysins for which structural information is available [22]. LysK is composed of the catalytic domains, CHAP (cysteine, histidine-dependent amidohydrolase/peptidase) and amidase-2 (*N*-acetylmuramoyl-L-alanine amidase), and the cell-wall-binding domain, SHb [22,23]. The catalytic domain hydrolyzes bacterial peptidoglycan, and the cell-wall-binding domain selectively binds to the target peptidoglycan [24]. LysK inhibits a wide range of staphylococci, and its synergistic antibacterial effect with a staphylococcal bacteriocin, lysostaphin, increases its activity against *S. aureus* [25].

Autolysins of bacteria are peptidoglycan hydrolases involved in cell wall maintenance, remodeling, growth, and separation during the life cycle and coordinate with various cell wall proteins associated with motility, adherence, nutrient transport, and virulence [26,27,28,29]. The catalytic domains of autolysins, such as *N*-acetylglucosaminidase, *N*-acetylmuramidase, N-acetylmuramyl-L-alanine amidase and endopeptidase, and carboxypeptidases [30], have different enzymatic functions, and the major autolysins of *S. aureus* are SsaA (Sle1), Atl, and LytH [31,32,33]. Endolysin secretory antigen SsaA-like protein (LysM peptidoglycan-binding domain-containing protein, SsaA-like protein) is modular and consists of cell-wall-binding (LysM) and catalytic (CHAP) domains. Fusion of the CHAP domain with the lysostaphin cell-wall-binding domain increases antibacterial activity [34]. To date, various chimeric lysins with different artificial combinations of functional domains of endolysins, autolysins, and bacteriocins have been tested for antibacterial activity [34,35,36,37]. However, the preparation of chimeric lysins for antibacterial activity screening is largely dependent on *E. coli* expression and purification, and more rapid and simpler process is required. Previously, we used a cell-free expression system for preparation of the novel chimeric lysin (gCHAP-LysK), combining the CHAP domain of SsaA-like protein, and amidase and the SH3b domains of LysK to observe antibacterial activity [38]. However, it was only applied to the colony reduction test which requires more experimental steps and time and is less suitable for multiple sample processing than the turbidity reduction test. In this study, we aimed at testing applicability of an improved cell-free expression system for screening and evaluation of chimeric lysin candidates using the turbidity reduction test. We demonstrated applicability of the cell-free expression system for rapid screening and comparison of antibacterial activity of chimeric lysin candidates and verified the antibacterial activity of ALS2 against the major *S. aureus* genotype causing bovine mastitis in Korea. Additionally, we demonstrated the presence of an unknown resistance mechanism employed by major *S. aureus* genotypes of humans and chickens.

## 2. Results

### 2.1. Rapid Antibacterial Activity Screening and Comparison of Chimeric Lysin Candidates

Chimeric lysin candidates were created through fusion with various enzymatic and cell-wall-binding domains based on LysK lysin, autolysins and lysostaphin as shown in Figure 1 and Appendix A. DNA templates for the chimeric lysins were prepared, and cell-free expression and antibacterial activity tests were performed (Figure 2). Due to the prevalence among bovine mastitis and human infection, RST10-2 and RST4-1, respectively, were included in the six strains of antibacterial activity screening panel comprising PMB4-1 (RST22-1), PMB8-1 (RST8-1), and PMB64-1 (RST10-2) from bovine mastitis; CCARM3806 (RST10-2) and CCARM3837 (RST4-1) from human infections; and BP11069 (RST6-3) from chicken infection (Table 1). In contrast to other chimeric lysins, ALS2 and ALS10 showed significantly different antibacterial activity from the negative control. In particular, ALS2 showed antibacterial activity higher than for ALS10 but much lower than the positive control, lysostaphin. The values for turbidity ratio (TR) of ALS2 (OD of ALS2 to OD of negative control at 6 h post treatment (hpt)) were different in several strains: 0.20 (PMB4-1), 0.23 (PMB8-1), 0.40 (PMB64-1), 0.57 (CCARM3806), 0.72 (CCARM3837), and 0.60 (BP11069) (Figure 3), and the various *S. aureus* strains exhibited differing susceptibilities.

To test the negative effects of the putative transmembrane (TM) regions of ALS5 (2–23 amino acid) and ALS6 (2–36 amino acid), ALS5-dTM and ALS6-dTM were constructed and tested for their antibacterial activity. In contrast to ALS6-dTM, ALS5-dTM showed significantly lower OD values at 4, 5, and 6 hpt, and the removal of transmembrane peptides increased the antibacterial activity. However, ALS5-dTM showed much lower antibacterial activity than ALS2 (Appendix A).

We compared the antibacterial activity of ALS2 to those of gCHAP and ALS2_Lyso to understand the synergistic roles of amidase and SH3 domains of LysK, and SH3 domain of lysostaphin with gCHAP (Appendix A). They showed less antibacterial activity than ALS2 and did not have increased antibacterial activity against less susceptible strains. Therefore, the combination of gCHAP with amidase and the SH3 domains of LysK increased antibacterial activity, and the SH3 domain may be not associated with the different susceptibilities of *S. aureus* strains to ALS2.

### 2.2. Comparison of Susceptibility of Various S. aureus Strains to ALS2 Using Cell-Free Expression System

Due to the observed differences in susceptibility to ALS2, we increased the number of *S. aureus* strains to 37 and tested their susceptibility (Table 1). All the tested strains showed 0.2–0.96 of TR and distinct susceptibilities from each other. PMB4-1 (RST22-1), PMB8-1 (RST8-1), and PMB81-1 (RST10-2) were more susceptible than the others from bovine mastitis, and CCARM3806 (RST10-2) and BP11069 (RST6-3) were more susceptible than other human and chicken strains, respectively (Figure 3).

### 2.3. Optimization of E. coli Expression of ALS2

The *E. coli*-expressed ALS2 showed bands of the correct (52.4 kDa) and unexpected (43.9 kDa) size after Western blotting with anti-6 histidine antibody (Figure 4). The intensity ratio (IR) of subprotein to correct bands was significantly high, and we therefore tried to reduce expression of the subprotein. A new inframe open reading frame was predicted by the ORF finder program with methionine 89 representing the putative start codon. When the 88th codon coding glutamate was changed to a stop codon (GAA88TAA), only the subprotein band with no antibacterial activity was detected. To confirm the start site and to reduce production of the subprotein, we introduced missense mutations into the ATG codon at position 89 and expressed the resulting protein in *E. coli* for Western blotting. The missense mutations, CTG(L), ACG(T), and GGG(G), decreased the IR values from 0.74 (wild-type) to 0.08, 0.09, and 0.08, respectively, and translation initiation was significantly inhibited by these mutations. Unexpectedly, the missense mutations ATT(I) and ATA(I) resulted in an increase in IR values to 2.19 and 1.74, respectively, and even facilitated subprotein production (Appendix A). We tested the effects of CTG(L), ACG(T), and GGG(G) mutations on the antibacterial activity using the cell-free expression system, demonstrating they all lost antibacterial activity (Appendix A).

To reduce the subprotein without functional loss of ALS2, synonymous point mutations were introduced to the second transcription start sites (TSS), −10 (GAT81GAC), −34 (GCT73GCC), and −31 (TAT74TAC). The single (GAT81GAC) and triple (GAT81GAC, GCT73GCC and TAT74TAC) mutations significantly decreased the expression level of the subprotein (Figure 4A). The mutated ALS2 gene containing the triple mutations and ALS1 genes were expressed in *E. coli* and purified by Ni-NTA affinity chromatography and gel filtration (Figure 4B). The total amounts of ALS2 and ALS1 purified per volume of cell culture were 2.711 and 1.258 mg/L, respectively. All the purified proteins were aliquoted and stored at −70 °C until use, and ALS1 was observed to degrade more rapidly into multiple bands than ALS2 over time.

### 2.4. Comparison of Susceptibility of Bovine and Human S. aureus Strains to E. coli-Expressed and Highly Purified ALS2

Usually, *S. aureus*-infected chickens are culled without treatment, and we tested the antibacterial activity of the purified ALS2 against representative bovine and human *S. aureus* strains using plate lysis and turbidity reduction tests. In the plate lysis test, the tested strains showed different lysis zones depending on the concentrations of loaded ALS2 (10 µg, 1 µg, and 0.1 µg; Table 1 and Figure 5). In contrast to others, PMB66-1 (RST5-2), PMB208-2 (RST11-6), CCARM3805 (RST2-1), and CCARM3837 (RST4-1) were resistant to 10 µg of ALS2. In the turbidity reduction test, all the tested strains showed TR values ranging from 0.19 to 0.93, which are lower than 1, and their growth was inhibited to different extents (Figure 5). To match the results of plate lysis (10 µg) and turbidity reduction tests, we determined the optimal reference point of TR, with 0.5 showing the highest matching percent (89%, 17/19). This means that most of tested strains showing >0.5 and ≤0.5 TR may show negative and positive lysis results, respectively. The two mismatching strains were PMB67-1 (RST5-2) and PMB179-1 (RST4-1). PMB67-1 and PMB179-1 showed a lysis zone at 10 µg, but their TR was higher than 0.5 (Table 1 and Figure 5).

### 2.5. Accordance of Bacterial Susceptibility to ALS2 Expressed in Cell-Free and E. coli Expression Systems

We also determined the optimal reference point of TR that showed the highest correlation for susceptibility when comparing the results of the cell-free and plate lysis assays. For the determined reference point of TR, 0.7, the matching rate was 79% (15/19), and the four mismatching strains were PMB2-1 (RST10-2), PMB179-1 (RST4-1), PMB188-1 (RST10-3), and PMB232-1 (RST10-2). The matching rate between cell-free and *E. coli* expression results was 83% (19/23). PMB2-1, PMB188-1, and PMB232-1 were divided into less and more susceptible strains according to the cell-free and *E. coli* expression systems. Additionally, PMB179-1 was divided into more and less susceptible strains according to the cell-free and *E. coli* expression systems (Table 1, Figure 3 and Figure 5).

## 3. Discussion

The difference between the previous and present cell-free expression protocols is in the noncoding region of DNA template: the absence and presence of upstream AT-rich region from promoter and the length of 5′-end of transcript to the ribosome binding site (RBS), 16 and 52 nucleotides, respectively. Only the upstream (−17) AT-rich region is known to increase transcription of T7 RNA polymerase by increasing binding affinity to the promoter and the efficiency of promoter clearance [40], and the presence of the AT-rich region in the present protocol may affect increased transcription of chimeric lysin genes. Relatively high correlation between ALS2 expressed by the present cell-free system and *E. coli* in terms of antibacterial spectrum may support its usefulness as is (Appendix A). However, our cell-free expression system may be improved further by adding T7 terminator or other artificial homologues to the 3′-end of the DNA template for efficient transcription termination and by optimizing downstream box for efficient translation initiation [41,42]. An additional difference is in antibacterial activity testing. Instead of a colony reduction test, we used a turbidity reduction test, which is more reproducible and suitable for rapid and multiple screenings of chimeric lysin candidates due to saving materials, efforts, and time for colony counting.

Although some chimeric lysins negatively affected by pH, concentration of salt, and cofactor of cell-free expression system can be ruled out, the cell-free expression system may be useful to rapidly screen and compare antibacterial activity of chimeric lysin candidates. The presence of gCHAP guarantees detectable antibacterial activity, but the combination partners affected their relative activity. The addition of LysM domains of SsaA-like protein to ALS2 reduced the antibacterial activity of ALS10 in comparison with ALS2 (Figure 2). The antibacterial activity of ALS6 was only detected after deletion of transmembrane peptides (ALS6-dTM) (Appendix A). The gCHAP domain in combination with the amidase and SH3 domains of LysK showed superior antibacterial activity to the codon-optimized LysK (ALS1), and artificial replacement of CHAP(ϕK) with gCHAP was successful (Figure 2). The combination of gCHAP with SH3 domain of lysostaphin was reported to confer more antibacterial activity than the natural SsaA-like protein [34]. The antibacterial activity of CHAP domain is much stronger than intact LysK, and the amidase domain of LysK is suspected to possess antibacterial activity [43]. As in the previous report, the subprotein containing amidase and the SH3 domains of LysK did not show detectable antibacterial activity. In this study, gCHAP and ALS2_Lyso showed lower activity than ALS2, and the increased antibacterial activity of ALS2 may support synergistic roles of amidase and SH3 domains of LysK (Appendix A). Therefore, efforts to develop chimeric lysins composed of artificially combined domains of natural lysins for increased activity are valuable, and the burden of *E. coli* expression and purification of the non-toxic, and the risk of expression failure of toxic chimeric lysin candidates can be lessened via our cell-free expression system.

Cases of subprotein generation during the expression of foreign genes in *E. coli* are not rare. The key role of methionine as the starting amino acid has been taken for granted, but other amino acids may also play a role as the starting amino acid for protein synthesis in *E. coli* [44]. The apparently reduction in subprotein levels resulting from mutations M89L/T/G may be useful as information in solving similar problems in other cases, whereas the increased translation of subprotein after M89I mutation was totally unexpected (Appendix A). The exact roles of the 89M in gCHAP domain function and conformation have been unknown, but the fact that M89L/T/G abolished the antibacterial activity of ALS2 may reflect the importance of 89M. The 3D structure of CHAP domain of LysK has been reported, and conserved and variable amino acids of CHAP domains of bacteriophage lysins have been compared [45,46]. In this study, we compared the 3D structure of gCHAP in SsaA-like protein (AF-A0A6B5FVD2-F1-model_v3) with CHAPk. The proposed catalytic triad of CHAPk is composed of 54C, 117H, and 134E, and the corresponding amino acids of gCHAP are 16C, 71H, and 88E, respectively. The side chain of 89M next to 88E protrudes into the different side from catalytic triad, and it neighbors with the side chains of 18W, 19Y, and 96Y (Appendix A). Therefore, the mutations M89L/T/G may have indirectly affected the structural fitness of the catalytic triad for enzyme activity. Thus, the effects of various mutations on antibacterial activity of chimeric lysins could be rapidly determined using the present cell-free expression system.

The broad antibacterial spectrum of lysostaphin to Korean *S. aureus* strains may also support its clinical usage for bovine mastitis, but recently-developed potent chimeric lysins need to be compared in further study [47,48]. However, the differing susceptibilities of *S. aureus* strains to ALS2 may reflect the presence of a resistance mechanism against lysins. Our observation that lysostaphin in the SH3 domain did not increase antibacterial activity of ALS2_Lyso against less susceptible *S. aureus* strains may support the postulation that the differences in susceptibility may be determined by the gCHAP domain. The SH3 domain of lysostaphin binds to the stem peptide of peptidoglycan with relatively high affinity, and there are no reports on modification of the stem peptide in *S. aureus* [49,50]. The prevalence of less susceptible RST2-1 and RST4-1 genotypes in humans may be the result of the acquisition of resistance mechanisms to bacteriophage lysins, including ALS2 [6,13]. The function of SsaA-like protein is unclear, but it is shared by all *S. aureus* strains. Although several missense mutations in gCHAP domains are present, they are randomly distributed among susceptible and less susceptible strains. For this reason, they may be unrelated to the differing susceptibilities of *S. aureus* strains. Although the extracted peptidoglycan is susceptible, live bacteria is resistant to lysins [51]. Therefore, various resistance mechanisms affecting the amounts of *O*- and *N*-acetyl groups and cross-linking peptides, the teichoic acid content, and regulatory molecules may play a role in determining the susceptibility of live bacteria [52,53].

The relatively short shelf-life of lysostaphin and LysK-like lysin activities have been reported, but the exact reason for the unstable antibacterial activity is unclear. In this study, we observed degradation of highly purified ALS1 and ALS2 into small fragments over time. The enzymatic recognition sites of CHAP and amidase domains are the linkages of d-alanine-l-glycine and *N*-acetyl muramyl-l-alanine, respectively, and autolysis of ALS1 and ALS2 was unexpected [54]. Efforts to increase the shelf-life of lysins have been indirect, e.g., optimization of storage temperature or addition of stabilizer during lyophilization [55,56]. Understanding the reasons for autolysis may be useful for extending the shelf-life of various chimeric lysins and thus promote their clinical application.

In conclusion, our cell-free expression system was successfully applied in screening chimeric lysin candidates and comparing their activities and the effects of mutations. The less susceptible *S. aureus* strains may be useful in panels for screening of new chimeric lysins for antibacterial activity and characterizing the different susceptibilities of *S. aureus* strains to lysins encourage further study on resistance mechanisms. In addition, our observation supporting autolysis of purified lysins may provide clues useful to the efforts aimed at expanding the short shelf-life of lysins and promoting clinical applications of various chimeric lysins.

## 4. Materials and Methods

### 4.1. Bacterial Strains

The thirty-seven *S. aureus* strains used in the study are listed in Table 1. Eighteen and five strains were isolated from bovine mastitis and human infection cases in Korea and reported previously [39]. Of the 14 strains isolated from chickens suffering from arthritis or septicemia, 9 strains were previously reported and 5 strains were characterized in this study according to previously reported methods [11]. All strains were grown in tryptic soy agar and broth at 37 °C overnight with shaking at 200 rpm and stored at −70 °C until use.

### 4.2. Bioinformatics

The functional domains of endolysins and autolysins were predicted using Protein Database from NCBI (https://www.ncbi.nlm.nih.gov/protein, accessed on 15 January 2018). Transmembrane domain prediction was carried out using TMHMM-2.0 (https://services.healthtech.dtu.dk/service.php?TMHMM-2.0, accessed on 26 April 2022). The ORF and promoter regions of *E. coli* were predicted by BPROM, a promoter prediction tool (http://www.softberry.com/berry.phtml?topic=bprom&group=programs&subgroup=gfindb, accessed on 20 March 2020). The intensity of protein bands was measured using Image J (http://imagej.nih.gov/ij, accessed on 16 August 2022). The 3D structure of LysM peptidoglycan-binding domain-containing protein was obtained from and the locations and interaction of amino acid residues were visualized with 3D viewer in AlphaFold protein structure database (https://alphafold.ebi.ac.uk/, accessed on 1 November 2022) and UCSF chimera program [57].

### 4.3. Design and Construction of Chimeric Lysin Gene for Cell-Free Expression

We previously reported the genome sequences of ALS1, ALS2, and ALS3 [38]. Briefly, ALS1 is a codon-optimized bacteriophage K lysin (codop-LysK) consisting of CHAP(ϕK)-Amidase(ϕK)-SH3(ϕK) domains; ALS2 is a fusion gene composed of the CHAP domain (gCHAP) of SsaA-like protein and amidase(ϕK) and SH3(ϕK) domains of codop-LysK; ALS3 is a fusion gene of the amidase domain of autolysin (Atl_1) and CHAP (ϕK) and SH3 (ϕK) of LysK. ALS4, 5, and 10 are generated by the fusion of LytM (CHAP-peptidase), LytD and SsaA-like protein genes with amidase(ϕK) and SH3(ϕK) of LysK, respectively. ALS 6, 7, 8, and 9 are generated by the fusion of LytH (SH3-AmiC) with gCHAP, LytD, LytM, and SsaA-like protein genes, respectively. The full length of the lysostaphin gene was codon-optimized and synthesized as previously reported [58], and only the functional region (142–387) was amplified. The structure of chimeric lysins and reference genes of each domain were summarized in Appendix A and Figure 1. Each domain for fusion was amplified using the primers listed in Appendix A and ligated by splicing using overlap extension PCR (SOE-PCR, the mixture was composed 10 μL of 10× buffer (1 μL), dNTPs (2.5 mM, 1 μL) TaKaRa Ex Taq DNA polymerase (5 U/μL, 0.1 μL) distilled water (6.9 μL), and template DNA of each (50 ng/μL, 1.5 μL). The mixture was incubated for 10 cycles of 95 °C for 45 s; 58 °C for 45 s; 72 °C for 2 min) [59] to clone into TA vector.

The gene fragment (T7-RBS (88bp): 5′-gcgaaatTAATACGACTCACTATAGGGAGACACACGACGGTTTCCCTCTAGAAATAATTTTGTTTAACTTTAAGAAGGAGataccctt-3′) containing the T7 promoter and RBS (Appendix A) was amplified using T7-F and chimeric lysin-specific reverse (T7-phiKCHAP-R for ALS1 and 3; T7-gCHAP-R for ALS2; T7-LytM-R for ALS4; T7-LytD-R for ALS5; T7-LytH-R for ALS6, 7 and 8; T7-SsaA-R for ALS9 and 10) primers and the pEXP5-CT/TOPO vector as a template (Appendix A, STEP 1). Chimeric lysin genes were amplified with combinations of forward primers (T7-phiKCHAP-F, T7-gCHAP-F, T7-LytM-F, T7-LytD-F, T7-LytH-F, T7-SsaA-F) and reverse primers (phiKL-TAG-R, gCHAP-TAG-R, LytD-TAG-R, LytM-TAG-R and LytH-TAG-R). The mixture was composed 50 μL of 5× Phusion^TM^ HF buffer (10 μL), dNTPs (10 mM, 1 μL), Phusion DNA polymerase (2 U/μL, 0.5 μL), forward and reverse primers (1.25 μM, 1 μL of each), distilled water (35.5 μL), and template plasmid (6 ng/μL, 1 μL). The mixture was incubated at 98 °C for 30 s; 30 cycles of 98 °C for 10 s–52 °C for 30 s–72 °C for 30 s with a final extension step at 72 °C for 5 min) and purified using the MG Gel Extraction Kit (MGMED, Inc., Seoul, Republic of Korea) according to the manufacturer’s protocol. Purified individual T7-RBS and chimeric lysin amplicons were ligated by SOE-PCR (STEP 2), and the final PCR (STEP 3) was performed using a consensus T7-F primer and reverse primers containing TAG sequences for each chimeric lysin. The final PCR amplicons were purified using the PureLink™ PCR Purification Kit (Thermo Fisher Scientific Inc., Waltham, MA, USA) according to the manufacturer’s protocol.

### 4.4. Site-Directed Mutagenesis

Point mutations were performed to reduce the expression level of the ALS2 subprotein. Single and triple point mutations, GAA88TAA and GCT73GCC/TAT74TAC/GAT81GAC, respectively, were introduced in the putative promoter sequence using Muta-Direct™ (iNtRON Biotechnology, Seongnam, Republic of Korea), primer sets were listed in Appendix A. In addition, the start codon of the subprotein, the 89th codon (ATG) was mutated to other codons coding different amino acids such as ATC (Ile), CTG (Leu), TTG (Leu), GTG (Val), ATT (Ile), ATA (Ile), TGT (Cys), TGC (Cys), ACT (Thr), ACC (Thr), ACA (Thr), ACG (Thr), GGG (Gly), and GGC (Gly) (Appendix A). The primer sets used are listed in Appendix A.

### 4.5. Cell-Free Protein Synthesis

Cell-free protein synthesis using a DNA template was performed using a cell-free protein synthesis system (Thermo Fisher Scientific Inc.) according to the manufacturer’s protocol. Briefly, *E. coli slyD*-Extract (20 µL), 2.5× IVPS *E. coli* reaction buffer (-AA) (20 µL), 50 mM amino acid (-met) (1.25 µL), 75 mM methionine (1 µL), T7 Enzyme Mix (1 μL), DNA template (1 μg), and DNase-free distilled water were mixed and incubated for 30 min at 30 °C in a shaking incubator at 300 rpm. An amount of 2 X IVPS feeding buffer (25 μL), 50 mM amino acid (-met) (1.25 μL), 75 mM methionine (1 μL), and DNase-free distilled water (22.75 μL) were added, and the mixture incubated at 30 °C in a 300 rpm shaking incubator for 5 h 30 min. After the reaction was completed, it was placed on ice for 5 min and stored at 4 °C. Lysostaphin gene and sterilized distilled deionized water were added as positive and negative controls, respectively [58].

### 4.6. Antibacterial Activity Tests of Chimeric Lysins

Screening of antibacterial activity of proteins produced by cell-free protein synthesis was conducted as previously described with modifications. The PMB strains were diluted in TSB to 1 × 10^7^ colony-forming units (CFU)/mL. Bacteria (100 μL) and 10 μL of protein were mixed in a 96-well plate. The culture was grown for 6 h with shaking at 200 rpm in a 37 °C incubator, and the OD_600_ value was recorded every hour. Turbidity reduction assays were used in testing as previously described with minor modifications [25,60]. Briefly, *S. aureus* was grown to mid-log phage (OD_600_ = 0.5) and then centrifuged at 3000× *g* for 10 min. After resuspending the cells in protein buffer until OD_600_ = 1.0, 200 µL was used in each well of a 96-well plate. Protein (10 µL) containing a concentration of 10 µg was mixed in the 96-well plate, and the OD value was measured after 10 min. Controls contained only protein storage buffer instead of purified protein. The plate lysis test was tested by modifying a previous method [54]. In brief, bacteria cells were grown to mid-log phage (OD_600_ = 0.4~0.6) at 37 °C in TSB. The *S. aureus* cells spread on tryptic soy agar (TSA) plates and air-dried for 10 min. Purified protein was diluted by concentration (10 µg, 1 µg, 0.1 µg) in 10 µL of protein buffer and spotted onto a fresh lawn. The plate was air-dried for 10 min and incubated at 37 °C for 20 h.

### 4.7. E. coli Expression and Protein Purification

ALS2 and ALS1 genes were cloned into the pET101 expression vector using the Champion™ pET101 Directional TOPO™ Expression Kit (Thermo Fisher Scientific Inc.) and transformed into a One Shot TOP10 competent cell. The chimeric lysins cloned into the pET101 vector were expressed in *E. coli* (BL21(*DE3*)) as per the manufacturer’s protocol. We induced the proteins at 0.1 mM IPTG and with incubation at 20 °C for 20 h. Induced *E. coli* was washed with sterilized distilled water (DW) and resuspended in lysis buffer (50 mM NaH_2_PO_4_, 300 mM NaCl, 10 mM imidazole, pH 8.0) and sonicated. After centrifugation at 3000× *g* for 30 min at 4 °C, the supernatant was mixed with Ni-NTA resin for protein purification. After 3 times washing with native washing buffer (50 mM NaH_2_PO_4_, 300 mM NaCl, 20 mM imidazole, pH 8.0), the resin-bound recombinant proteins were eluted using native elution buffer (50 mM NaH_2_PO_4_, 300 mM NaCl, 250 mM imidazole, pH 8.0). After removing the side protein by size exclusion chromatography, it was exchanged with a protein storage buffer (400 mM NaCl, 20 mM Tris, 1% glycerol, pH 7.5). The protein content was measured by the Bradford assay, and the final protein was observed by SDS-PAGE.

### 4.8. SDS-PAGE and Western Blotting

Aliquots of proteins were electrophoresed on NuPAGE Bis-Tris protein gels (4–10%, Thermo Fisher Scientific Inc.). Proteins were transferred to Immobilon-P membranes (Thermo Fisher Scientific Inc.) and blocked with 5% (*w*/*v*) skim milk in TBS (milk/TBS) for 1 h. After washing 3 times with TBS, the membrane was incubated with HRP-conjugated anti-6 histidine goat polyclonal antibody (A190-113P; Bethyl Laboratories, Montgomery, TX, USA) diluted 3000-fold in 1% skim milk. After washing the membrane, it was developed with TMB One Component HRP Membrane Substrate (Surmodics, Inc., Eden Prairie, MN, USA) according to the manufacturer’s instructions. The sodium dodecyl sulfate polyacrylamide gel electrophoresis (SDS-PAGE) gel was stained with Coomassie Brilliant Blue R-250 (Biosesang, Seongnam, Republic of Korea).

### 4.9. Statistical Analyses

The statistical significance of antibacterial activity was assessed using *t*-test and one-way analysis of variance (ANOVA) followed by Dunnett’s test SPSS Statistics Windows, ver. 26.0. *p*-values less than 0.05, 0.01, and 0.001 were considered as significant. All experiments were performed in triplicate for each sample. Triplicated and duplicated independent experiments were indicated.

## Figures and Tables

**Figure 1 antibiotics-12-00667-f001:**
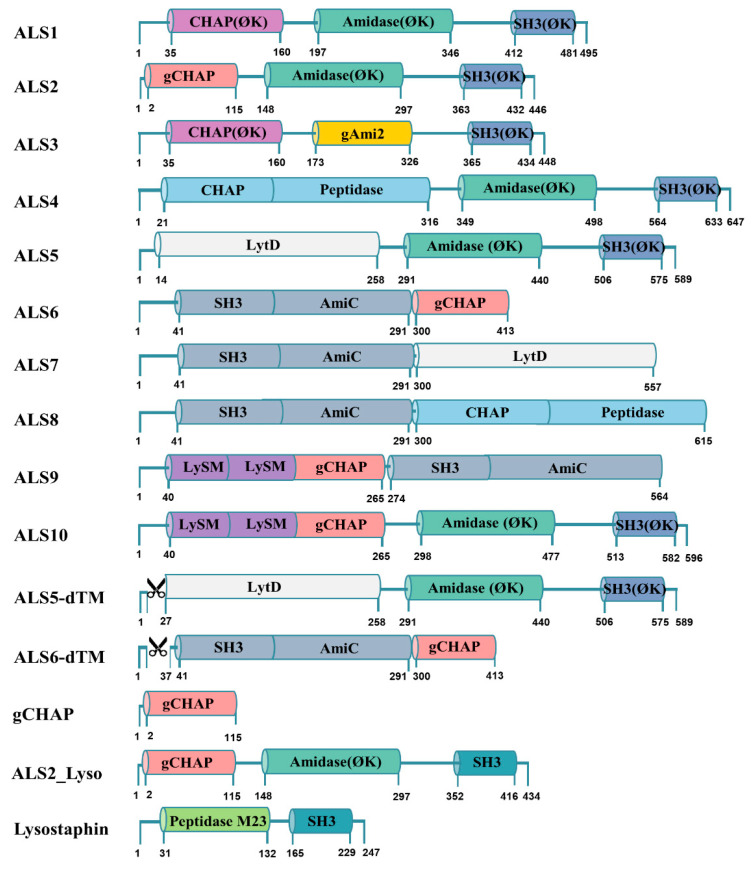
Construction of chimeric lysins. Colors that are common indicate the same domain. The number under the domain indicates the position of the amino acid of the chimeric lysin. ⌀K indicates codon-optimized LysK (AFN38929); gCHAP domain of ALS2, 6, 9, 10, ALS6-dTM, gCHAP, and ALS2_Lyso (152–265 of SsaA-like protein:); gAmi2 domain of ALS3 (257–385 of autolysin CwlA:); CHAP-Peptidase domain of ALS4, 8 (1–316 of glycine-glycine endopeptidase, LytM:); LytD domain of ALS5,7 (1–258 of N-acetylglucosaminidase:); SH3-AmiC domain of ALS6, 7, 8, 9 (1–291 of cell wall amidase, LytH); LysM-LysM-gCHAP domain of ALS9,10 (1–265 of SsaA-like protein). Lysostaphin (144–389), composed of peptidase M23 domain and SH3 domain, is codon-optimized. Additionally, the transmembrane regions of ALS5 (2–26) and ALS6 (2–36) were deleted. In ALS2_Lyso, the gCHAP and amidase domains of ALS2 (1–351) and the SH3 domain of lysostaphin (352–434) are fused.

**Figure 2 antibiotics-12-00667-f002:**
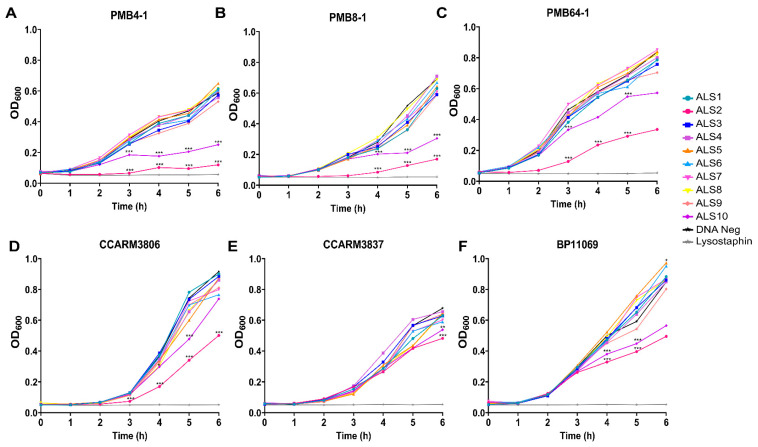
Screening of antibacterial activity of chimeric lysin candidates. Chimeric lysin candidates were prepared using cell-free expression system and antibacterial activity was tested using turbidity reduction test. Antibacterial activities of chimeric lysin candidates were tested on *S. aureus* strains from bovine mastitis: (**A**) PMB4-1, (**B**) PMB8-1, and (**C**) PMB64-1; from human infection: (**D**) CCARM3806 and (**E**) CCARM3837; and from chicken infection (**F**) BP11069 for 6 h. Cell-free expression reaction with lysostaphin DNA template without DNA template were used as positive and negative controls, respectively. All experiments were performed in triplicate samples and independent experiments were duplicated for PMB8-1 and CCARM3806. Single experiment was performed for other strains. The median values are shown. Data were analyzed by one-way ANOVA followed by Dunnett’s test to determine the significance relative to the negative control. (*** *p* < 0.001, ** *p* < 0.01, * *p* < 0.05).

**Figure 3 antibiotics-12-00667-f003:**
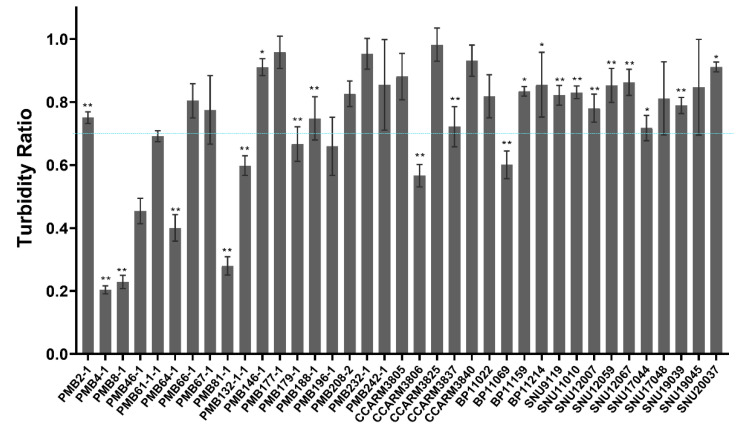
Different susceptibility of *S. aureus* strains to ALS2. After 6 h incubation of bacteria and ALS2, the turbidity ratio (TR) for each strain was determined by calculating the OD_600_ value ratio of ALS2-treated to untreated samples. All experiments were performed in triplicate samples and independent experiments were triplicated for PMB2-1, PMB4-1, PMB8-1, PMB64-1, PMB67-1, PMB179-1, and CCARM3806, and duplicated for PMB66-1, PMB196-1, CCARM3837, CCARM3840, and BP11069. Single experiment was performed for other strains. The mean ± standard deviation of TR values is presented. In total, 37 strains from bovine mastitis (18 PMB strains), human infection (5 CCARM strains), and chicken infection (14 BP/SNU strains) were compared. The broken line represents the reference TR (0.7). The significance of the OD values of ALS2 treated and untreated strain was analyzed by *t*-test. (*n* = 3, ** *p* < 0.01, * *p* < 0.05).

**Figure 4 antibiotics-12-00667-f004:**
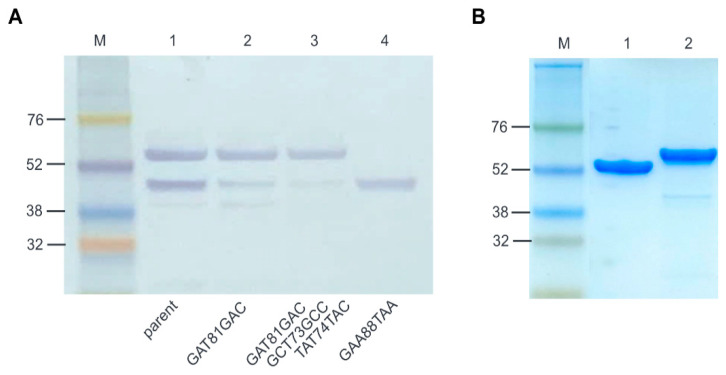
Reduced subprotein by mutation of the innate promoter of ALS2 and purification of ALS2. (**A**) Western blotting of ALS2 expressed from parent and mutated DNA templates. The molecular weight of the 6 histidine-tagged ALS2 is approximately 52.4 kDa and the subprotein is approximately 42.9 kDa after *E. coli* expression. Lane M: protein molecular weight marker (Amersham™ ECL™ Rainbow™ Marker, Lane 1: ALS2; Lane 2: ALS2 expressed from ALS2-TS-10 gene with GAT81GAC silent mutation; Lane 3: ALS2 to expressed from ALS2-TS-31 gene with GAT81GAC, GCT73GCC and TAT74TAC silent mutations; Lane 4: Subprotein expressed from ALS2-Stop gene with GAA88TAA nonsense mutation. (**B**) SDS-PAGE of purified ALS2 and ALS1 using 4–10% gradient polyacrylamide gel. Lane M: protein molecular weight marker; Lane 1: ALS2; Lane 2: ALS1.

**Figure 5 antibiotics-12-00667-f005:**
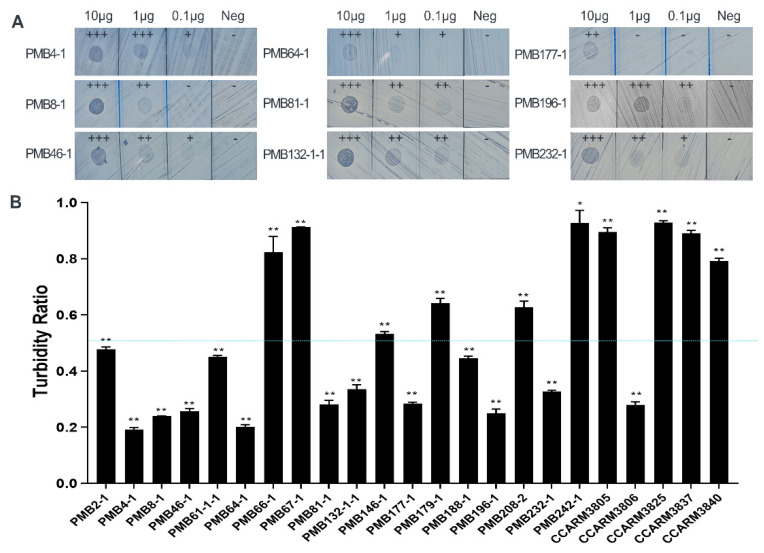
Comparison of antibacterial activities of *E. coli*-expressed and highly purified ALS2 using plate lysis and turbidity reduction tests. (**A**) Plate lysis test of purified ALS2. Treatment with the protein resulted in formation of a clear zone against various PMB strains (isolated from bovine) at concentrations of 10/1/0.1 μg/μL for the negative control (Neg), only protein purification buffer was used. +++, completely clear lysis; ++, incompletely clear lysis; +, distinguishable lysis in comparison with negative; -, no lysis. (**B**) After 10 min incubation of bacteria and highly purified ALS2, the turbidity ratio (TR) for each strain was determined by calculating the OD_600_ value ratio of ALS2-treated to untreated samples. Single experiment was performed in triplicate samples and the mean ± standard deviation of TR values is presented. The significance of the ALS2 treated and untreated groups was analyzed by *t*-test. (*n* = 3, ** *p* < 0.01, * *p* < 0.05). A total of 23 strains of *S. aureus* from bovine mastitis (18 PMB strains) and human infection (5 CCARM strains) were compared. The broken line represents the reference TR (0.5).

**Table 1 antibiotics-12-00667-t001:** *S. aureus* strains used in the study and their susceptibility to ALS2.

Strains	Origin ^a^	RST ^b^	MRSA Gene	Plate Lysis Test ^c^			Turbidity Reduction Test		Ref
				10 µg	1 µg	0.1 µg	*E. coli* ^d^	Cell-Free ^e^	
**PMB2-1**	BM	10-2	-	++ ^c^	+	-	+	- ^f^	[14]
**PMB4-1**	BM	22-1	-	+++ ^b^	+++	+	+	+ ^f^	
**PMB8-1**	BM	8-1	-	+++	++	-	+	+ ^f^	
**PMB46-1**	BM	10-2	-	+++	++	+	+	+	
**PMB61-1-1**	BM	10-2	-	++	+	+	+	+	
**PMB64-1**	BM	10-2	-	+++	+++	++	+	+ ^f^	
**PMB66-1**	BM	14-3	-	-	-	-	-	- ^g^	
**PMB67-1**	BM	5-2	-	+	-	-	-	- ^f^	
**PMB81-1**	BM	10-2	-	+++	++	++	+	+	
**PMB132-1-1**	BM	10-2	-	+++	++	++	+	+	
**PMB146-1**	BM	11-5	-	+++	++	-	+	+	
**PMB177-1**	BM	14-2	-	++	+	-	+	+	
**PMB179-1**	BM	4-1	-	+	-	-	-	+ ^f^	
**PMB188-1**	BM	10-3	-	++	+	-	+	-	
**PMB196-1**	BM	2-1	-	+++	++	++	+	+ ^g^	
**PMB208-2**	BM	11-6	-	-	-	-	-	-	
**PMB232-1**	BM	10-2	-	+++	++	+	+	-	
**PMB242-1**	BM	11-4	-	nt ^f^	nt	nt	-	-	
**CCARM3805**	HI	2-1	+	-	-	-	-	-	
**CCARM3806**	HI	10-2	+	nt	nt	nt	+	+ ^f^	
**CCARM3825**	HI	2-1	+	nt	nt	nt	-	-	
**CCARM3837**	HI	4-1	+	-	-	-	-	- ^g^	
**CCARM3840**	HI	4-1	+	nt	nt	nt	-	- ^g^	
**BP11022**	CI	3-1	-	nt	nt	nt	nt	-	[13]
**BP11069**	CI	6-3	+	nt	nt	nt	nt	+ ^g^	
**BP11159**	CI	3-1	-	nt	nt	nt	nt	-	
**BP11214**	CI	3-1	-	nt	nt	nt	nt	-	
**SNU9119**	CI	3-1	-	nt	nt	nt	nt	-	
**SNU11010**	CI	3-1	-	nt	nt	nt	nt	-	
**SNU12007**	CI	3-1	-	nt	nt	nt	nt	-	
**SNU12059**	CI	3-1	-	nt	nt	nt	nt	-	
**SNU12067**	CI	3-1	-	nt	nt	nt	nt	-	
**SNU17044**	CI	3-1	-	nt	nt	nt	nt	-	This study
**SNU17048**	CI	3-1	-	nt	nt	nt	nt	-	
**SNU19039**	CI	3-1	-	nt	nt	nt	nt	-	
**SNU19045**	CI	3-1	-	nt	nt	nt	nt	-	
**SNU20037**	CI	3-1	-	nt	nt	nt	nt	-	

^a^ BM: bovine mastitis, HI: human infection, CI: chicken infection. ^b^ RST: *rpoB* sequence type [39]. ^c^ The susceptibility was graded as below: +++, completely clear lysis; ++, incompletely clear lysis; +, distinguishable lysis in comparison with negative; -, no lysis. ^d^ +, TR ≤ 0.5; -, TR > 0.5. ^e^ +, TR ≤ 0.7; -, TR > 0.7. ^f^ nt, not tested. ^f^ Independent experiment was triplicated and ^g^ duplicated.

## Data Availability

Data is contained within the article or Appendix A.

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
