# Peer review of "Rapid Screening and Comparison of Chimeric Lysins for Antibacterial Activity against Staphylococcus aureus Strains"

_antibiotics, 2023, doi:10.3390/antibiotics12040667_

Round 1

Reviewer 1 Report (New Reviewer)

The manuscript by Park et al. entitled  “Rapid Screening and Comparison of Chimeric Lysins for Antibacterial Activity against Staphylococcus aureus Strains” describes a cell-free expression system for screening chimeric lysin candidates and comparing their activities and the effects of mutations.

The paper is well-written with good background information. The methods used are described adequately and the authors present data which support their conclusions.

Following minor revision, this paper deserves publication in Antibiotics.

Comments

Abstract:

No abbreviations, spell out CHAP (cysteine, histidine-dependent amidohydrolase/peptidase)

Page 2: Gram-positive

References: Literature references should be updated. For example, the first 13 references contain only one recent from 2021, the rest are much older.

Author Response

Dear reviewer 1

Thank you for your review and comments. And we could improve our manuscript thank to your comments.

Comments

Abstract:

No abbreviations, spell out CHAP (cysteine, histidine-dependent amidohydrolase/peptidase)

>> As suggested, we have added an abbreviation to the abstract.

Page 2: Gram-positive

>> As per the advice, We have modified it to Gram-positive

References: Literature references should be updated. For example, the first 13 references contain only one recent from 2021, the rest are much older.

>> As per the advice, we replaced some old with recent references.

Reviewer 2 Report (New Reviewer)

This manuscript brings 2 main messages: regarding the rapid method for screening and the activity of chimeric lysins. This might make somehow the readers to be not focus. However, the authors wrap these idea into one and wrote the manuscript well. The experiments are also well designed. Nevertheless, there are some points need to be addressed to improve the manuscript:

- Table 1. The authors need to explain why some of the strains were not tested.

- Figure 2 resolution needs to be increased.

- Figure 3. It makes confusion the t-test is testing the bars shown compred to which bar. As the authors mentioned that they compared the untreated to treated it will be too much to put in the graph. I suggest to transfrom this graph into table since it has many strains to show as well.

- Figure 4. It will be easier for the reader if the authors provide the type of  mutations to be included in the figure.

- Figure 5b. same as figure 3.

Author Response

Dear reviewer 1

Thank you for your review and comments. And we could improve our manuscript thank to your comments.

This manuscript brings 2 main messages: regarding the rapid method for screening and the activity of chimeric lysins. This might make somehow the readers to be not focus. However, the authors wrap these idea into one and wrote the manuscript well. The experiments are also well designed. Nevertheless, there are some points need to be addressed to improve the manuscript:

- Table 1. The authors need to explain why some of the strains were not tested.

>> The amount of highly purified ALS2 (Ni-NTA column and gel-filtration column) was not enough and unstable. Therefore, we focused on representative strains of bovine mastitis and human infections requiring treatment. Usually S. aureus-infected chickens are culled without treatment.

- Figure 2 resolution needs to be increased.

>> Following the advice, we have increased the resolution of Figure 2.

- Figure 3. It makes confusion the t-test is testing the bars shown compared to which bar. As the authors mentioned that they compared the untreated to treated it will be too much to put in the graph. I suggest to transfrom this graph into table since it has many strains to show as well.

>> I’m sorry for the confusion. The results in the Figure 3 and 5 are the turbidity ratio of ALS2-treated and untreated sample (strain) and the significance tested between them was shown simply on the ratio bar. Therefore, we hope to maintain the figures.

- Figure 4. It will be easier for the reader if the authors provide the type of mutations to be included in the figure.

>> As recommended, we have included the point mutation type in Figure 4A.

- Figure 5b. same as figure 3.

>> Same as Figure 3.

Reviewer 3 Report (New Reviewer)

Authors designed chimeric lysins candidates based on cell wall-lysing and CWB domains of endolysins, autolysins, and bacteriocins, and constructed the cell free expression system. Active chimeric lysins candidates were rapidly screened and their activity were detected against s. aureus by using turbidity reduction test. ALS expressed in E. coli and displayed more potent activity against S. aureus only from bovine mastitis, but lower activity against S. aureus from human and chicken. This study indicated that ALS2 may be used as one of lead molecules. It is very interesting and important work. However, some questions need to resolve before publishing:

1、   Table 1: please add S. aureus in table title; are the strains from the same farm or different ones? What is relationship among the strains from three sources?

2、   The turbidity reduction test was used to detect the antibacterial activity of chimeric lysins candidates, do you determine the MICs or MBCs of ALS2?

3、    In Fig. 2, which is the blank control (without candidates)? Could some candidates improve bacterial growth?

4、    The expressed levels of ALS2 and ALS1 are low (final level of 2.711 and 1.258 mg/L), please explain the reasons. And what about their purity?

Author Response

Dear reviewer 3

Thank you for your revew and comments. We could improve our manuscript thanks to your comments.

Authors designed chimeric lysins candidates based on cell wall-lysing and CWB domains of endolysins, autolysins, and bacteriocins, and constructed the cell free expression system. Active chimeric lysins candidates were rapidly screened and their activity were detected against s. aureus by using turbidity reduction test. ALS expressed in E. coli and displayed more potent activity against S. aureus only from bovine mastitis, but lower activity against S. aureus from human and chicken. This study indicated that ALS2 may be used as one of lead molecules. It is very interesting and important work. However, some questions need to resolve before publishing:

  1. Table 1: please add S. aureus in table title; are the strains from the same farm or different ones? What is relationship among the strains from three sources?

>> As recommended we added ‘S. aureus’ in the Table title.

>> All the S. aureus strains from bovine mastitis and chicken arthritis cases were isolated from epizootically independent farms. Additionally, human strains from the Korean Antibiotic-Resistant Microbial Collection (CCARM) are unrelated each other.

2、The turbidity reduction test was used to detect the antibacterial activity of chimeric lysins candidates, do you determine the MICs or MBCs of ALS2?

>> We made an effort to determine the MIC of highly purified ALS2 protein (Ni-NTA and gel filtration column purification). However, as noted in the Discussion (page 11), due to autolysis of the ALS2 protein, we could not determine reproducible MIC. Therefore, we did not include the data.

  1. In Fig. 2, which is the blank control (without candidates)? Could some candidates improve bacterial growth?

    >> The blank control in Figure 2 is cell-free expression reaction added same volume of DNase-free water without DNA. As the cell-free expression reaction includes E. coli extract containing translation machinery, amino acids, enzymes, feed buffer, etc., and they may help bacterial growth to some extent.

4、The expressed levels of ALS2 and ALS1 are low (final level of 2.711 and 1.258 mg/L), please explain the reasons. And what about their purity?

>> Yes, it seems very low level. Considering other reports, the expression levels of chimeric lysins are very high. Usually they measure the amount of protein after Ni-NTA clolumn purification, but we measured the protein amount after two step purification of Ni-NTA and gel filtration columns. Therefore, the amount of highly purified proteins may be less. The purities of fractions collected during gel filtration were confirmed by SDS-PAGE and only single band fractions (E26-28) were pooled and used for further study.

Sincerely yours,

Hyuk-Joon Kwon, DVM, PhD, Professor

Department of Farm Animal Medicine, College of Veterinary Medicine,

Seoul National University

Round 2

Reviewer 2 Report (New Reviewer)

I thank the authors for addressing most of the points that I asked to be improved. Only one last point regarding table 1, it would be great if the authors include the reasing in the result part but with more scientific reason, not with the insufficient protein or reagent.

Author Response

Dear Reviewer 2

Thank you very much for your comment.

We explained why S. aureus strains of chicken were not tested for the experiment in the Result section 2.4. as below.

'Usually S. aureus-infected chickens are culled without treatment, and we tested the antibacterial activity of the purified ALS2 against representative bovine and human S. aureus strains using plate lysis and turbidity reduction tests.'

Sincerely yours

Hyuk-Joon Kwon DVM, PhD, Professor

Laboratory of Poultry Medicine, Department of Farm Animal Medicine, College of Veterinary Medicine, Seoul Natinal University.

This manuscript is a resubmission of an earlier submission. The following is a list of the peer review reports and author responses from that submission.

Round 1

Reviewer 1 Report

In this study, the authors report an improved version of combining cell-free expression system with a turbidity reduction test and its application for screening activities of ALS2 chimeric lysins. Although there may be certain values, it is hard to understand the objective of the study and follow the experimental design.

1. Turbidity reduction test is a common test used for measuring the activity of lysins. Because there may be a lot of proteins in the cell free expression system, the authors need to confirm if the cell free system would affect the activity of lysin measured using turbidity test by adding a purified lysin into the system. 

2. Why were ASL2 lysins chosen? From the results, it seems that lysins based on ASL domains showed much less activity compared to lysostaphin, not to mention other reported good lysins against S. aureus, such as ClyO and ClyC etc. Are there any special reasons for screening the chimeric lysins indicated in Figure 1?

3. The number of the isolates tested is small to give the conclusion that the lysins are more active to isolates from bovine mastitis than from human and chicken infections.

4. Fig.2: it is strange to see that OD600 is increasing with increase in time. Why?  Does it mean that the turbidity was increasing after interaction with the lysins?

5. Fig. 3: It is also strange to see TR was used to judge the susceptibility of the lysins to different isolates. Normally MIC or MBC is used.

6. Fig. 4: Fig. B is not necessary and should be deleted. 

Author Response

In this study, the authors report an improved version of combining cell-free expression system with a turbidity reduction test and its application for screening activities of ALS2 chimeric lysins. Although there may be certain values, it is hard to understand the objective of the study and follow the experimental design.

  1. Turbidity reduction test is a common test used for measuring the activity of lysins. Because there may be a lot of proteins in the cell free expression system, the authors need to confirm if the cell free system would affect the activity of lysin measured using turbidity test by adding a purified lysin into the system. 

> We agree to the reviewer’s opinion. Lysin activity is affected by reaction condition such as cofactors, pH, salt concentration etc., so it may be not optimal condition for some chimeric lysin candidates which may be ruled out for negative activity during screening. We described the limit of cell free expression system in page 10 (Discussion section lines 7-8)

  1. Why were ASL2 lysins chosen? From the results, it seems that lysins based on ASL domains showed much less activity compared to lysostaphin, not to mention other reported good lysins against S. aureus, such as ClyO and ClyC etc. Are there any special reasons for screening the chimeric lysins indicated in Figure 1?

> We started our research to screen chimeric lysin candidates (ALS1-10) using cell free and E. coli expression systems in 2010. Using previous cell free expression system, we could only find antibacterial activity of ALS2 using colony reduction test. One of the aims of this study is evaluation of the improved cell free expression system whether it can be applicable to turbidity reduction test or not, so we selected somewhat old chimeric lysins due to apparent contrast as well as availability of the templates. Using the improved cell free expression system (template structure) we could clearly see the antibacterial activity of ALS2 and ALS6-dTM. As a further study now we are introducing mutations to improve antibacterial activity and spectrum, and shelf life of ALS2. Additionally we are going to determine rank orders of antibacterial activity of reported lysins and chimeric lysins to verify our cell free expression system. According to reviewer’s comment we described one of recently reported advanced chimeric lysins generated by domain shuffling in page 10 lines 48-49.

  1. The number of the isolates tested is small to give the conclusion that the lysins are more active to isolates from bovine mastitis than from human and chicken infections.

> We agree to reviewer’s comment. We tested only several representative strains of RSTs prevalent among cows, chickens and humans. Previously, we observed the highest nucleotide identity between strains of same RSTs [e.g. RST10-2 strains of Korean cows, PMB64-1 (CP034486) vs. PMB81-4 (CP034441) 99.98%]. Therefore, we expect similar susceptibility of major RST10-2 strains in Korea [when we tested RST10-2 was positive in 27.3% farms (6/22), 37.7% cows (29/72) and 41.7% udders (35/84), Ko et al., 2019 Journal of Microbiology]. As the case of RST10-2 strains of cows the strains of RST4-1 and RST2-1 are most identical between strains of same RSTs in terms of genome structure and sequences (Ko et al., 2021 Res Vet Sci). In case of predominant RST3-1 chicken strains showed much less susceptibility than cow strains.

  1. Fig.2: it is strange to see that OD600 is increasing with increase in time. Why?  Does it mean that the turbidity was increasing after interaction with the lysins?

> S. aureus (1 × 107 CFU/mL) was incubated in TSB with and without chimeric lysins. The increase of OD600 reflect less antibacterial activity of chimeric lysins. Although lysostaphin showed no increase of OD600 ALS2 showed stronger antibacterial activity than other chimeric lysins with showing less steep increase of OD600.

  1. Fig. 3: It is also strange to see TR was used to judge the susceptibility of the lysins to different isolates. Normally MIC or MBC is used.

> We agree to reviewer’s comment that MIC and MBC is the golden standard to evaluate susceptibility of different isolates to certain antibiotics. But we intended to screen antibacterial activity of chimeric lysin candidates without E. coli expression and protein purification. If a new chimeric lysin candidate show significantly lower TR value to less susceptible isolates in our study it is valuable to select it for E. coli expression and protein purification to measure MIC and MBC.

  1. Fig. 4: Fig. B is not necessary and should be deleted. 

>> As per recommendation, figure 4.B was removed.

Reviewer 2 Report

The authors applied the cell-free expression system for rapid screening and comparison of the antibacterial activity of chimeric lysin candidates and verified the antibacterial activity of ALS2 against S. aureus. Please find my comments below:

1. Before abbreviating MLST and PFGE, please write the complete term and then abbreviate in the text. 

2. At the end of the introduction please clearly state your hypothesis for this study before stating the summary of your obtained results. 

3. I recommend increasing the font size for Figure 2 a bit more for better clarity of the legend.

4. Figure 2 - The authors state the experiments were done in triplicates. Could you please specify if these are technical replicates? How many times did you run the experiment with triplicates to reproduce your results? At least 3 times is standard. 

5. Figure 3- Please specify the statistical details in the figure legend. The figure legend is incomplete as it is. Also, please use the same font throughout the figures for consistency. 

6. Figure 5- Please specify the statistical details in the figure legend. The figure legend is incomplete as it is. Also, please use the same font throughout the figures for consistency. 

7. In the methods, please specify if the strains were grown with or without shaking. 

8. For the statistical analysis section specify the program you used for the analysis, as well as the p-value threshold. Also, include that the experiments were performed in triplicate and repeated "x" amounts of time for reproducibility of results. 

Author Response

The authors applied the cell-free expression system for rapid screening and comparison of the antibacterial activity of chimeric lysin candidates and verified the antibacterial activity of ALS2 against S. aureus. Please find my comments below:

  1. Before abbreviating MLST and PFGE, please write the complete term and then abbreviate in the text. 

> We revised as recommended.

  1. At the end of the introduction please clearly state your hypothesis for this study before stating the summary of your obtained results. 

> We added the aim of study at the end of the Introduction section.

  1. I recommend increasing the font size for Figure 2 a bit more for better clarity of the legend.

>> As per recommendation, the font for Figure 2 has been increased.

  1. Figure 2 - The authors state the experiments were done in triplicates. Could you please specify if these are technical replicates? How many times did you run the experiment with triplicates to reproduce your results? At least 3 times is standard. 

> All experiments were performed in triplicate for each sample (technical triplicates). Some of strains described in the legend of Figure 2 were repeated three times or two times by independent experiments but others were done by single experiment.

  1. Figure 3- Please specify the statistical details in the figure legend. The figure legend is incomplete as it is. Also, please use the same font throughout the figures for consistency. 

> We revised as recommended.

  1. Figure 5- Please specify the statistical details in the figure legend. The figure legend is incomplete as it is. Also, please use the same font throughout the figures for consistency. 

> We revised as recommended.

  1. In the methods, please specify if the strains were grown with or without shaking. 

> We revised as recommended.

  1. For the statistical analysis section specify the program you used for the analysis, as well as the p-value threshold. Also, include that the experiments were performed in triplicate and repeated "x" amounts of time for reproducibility of results. 

> We revised as recommended and explained in detail in Figure legends and Statistical analysis section. All experiments were performed in triplicate for each sample. Although we did not perform independent experiment in triplicate the susceptibility of S. aureus strains to ALS2 were reproduced by three times using turbidity reduction and plate lysis tests with cell free- and E. coli-expressed, and E. coli-expressed ALS2, respectively (Table 1). We indicated repeated numbers of independent experiments in Table 1. In case of Figure S2 and S3 single experiment was performed but the relative antibacterial activities of tested chimeric lysins were duplicated and triplicated using two and three different strains, respectively, in the end.

Round 2

Reviewer 1 Report

The revised manuscript has improved slightly compared to the previous version. From their responses to the previous comments, I understand better why the authors have done the current study. But it needs to clearly state the aims of the study is to evaluate if the improved cell free expression system could combine with turbidity reduction test to screen chimeric lysins. Then the experiments should be designed to show what are the differences in the cell free expression to make the improved systems work and not the previous system.  In the discussion, the authors should explain more on why the improved system is better than other screening system, such as in terms of efficiency, time, lysins toxic to E. coli, etc. 

Overall, I still felt that the revised manuscript is not clear on the purpose of the study and needs significant modifications in the experimental design to support the aim, i.e. prove its scientific value.

Author Response

Dear Reviewer

We revised our manuscript and supplementary materials as followings.

Thank you for your comments that apparently improved our manuscript.

The revised manuscript has improved slightly compared to the previous version. From their responses to the previous comments, I understand better why the authors have done the current study.

But it needs to clearly state the aims of the study is to evaluate if the improved cell free expression system could combine with turbidity reduction test to screen chimeric lysins.

> As recommended we described more in detail to clarify the aim of the present study in page 2 lines.

 Then the experiments should be designed to show what are the differences in the cell free expression to make the improved systems work and not the previous system. 

> We added panel A to compare the difference between previous and present cell free expression templates in Figure S1. We referred Figure S1 in page 9 in the Discussion section and page 12 in M&M section (subtitle 4.3.).

In the discussion, the authors should explain more on why the improved system is better than other screening system, such as in terms of efficiency, time, lysins toxic to E. coli, etc. 

> We added additional explanation as recommended in page 10 in lines 6-9, 29-30 briefly.